# Facilitating the dry reforming of methane with interfacial synergistic catalysis in an Ir@CeO$_{2-x}$ catalyst

Hui Wang[1], Guoqing Cui [2] ✉, Hao Lu[1], Zeyang Li[1], Lei Wang[1,3], Hao Meng [1,3], Jiong Li [4], Hong Yan[1], Yusen Yang [1,3] ✉ & Min Wei [1,3] ✉

The dry reforming of methane provides an attractive route to convert greenhouse gases (CH$_4$ and CO$_2$) into valuable syngas, so as to resolve the carbon cycle and environmental issues. However, the development of high-performance catalysts remains a huge challenge. Herein, we report a 0.6% Ir/CeO$_{2-x}$ catalyst with a metal-support interface structure which exhibits high CH$_4$ (~72%) and CO$_2$ (~82%) conversion and a CH$_4$ reaction rate of ~973 $\mu mol_{CH4}$ $g_{cat}^{-1} s^{-1}$ which is stable over 100 h at 700 °C. The performance of the catalyst is close to the state-of-the-art in this area of research. A combination of in situ spectroscopic characterization and theoretical calculations highlight the importance of the interfacial structure as an intrinsic active center to facilitate the CH$_4$ dissociation (the rate-determining step) and the CH$_2$* oxidation to CH$_2$O* without coke formation, which accounts for the long-term stability. The catalyst in this work has a potential application prospect in the field of high-value utilization of carbon resources.

Owing to the increasing global warming and climate change issues, strategies for greenhouse gas reduction have drawn extensive interest from both fundamental research and industrial applications[1–3]. CO$_2$ and CH$_4$ are regarded as two predominant contributors to the greenhouse effect; therefore, their utilization and conversion to high-value-added chemicals and fuels meet the demands for achieving large-scale carbon fixation, carbon emission reduction and carbon cycle[4–7]. One promising approach is to convert both CO$_2$ and CH$_4$ simultaneously through thermo-catalytic dry reforming of methane (DRM) reaction, which produces the syngas (H$_2$ and CO) as an important platform for alternatives of petroleum-derived fuels and valuable chemicals[8–11]. Thermodynamically, the DRM reaction involves both C−H bond dissociation (439 kJ mol$^{-1}$) and C=O bond hydrogenation (750 kJ mol$^{-1}$) followed by subsequent formation of CO and H$_2$, resulting in a highly endothermic process ($\Delta H_{298K}$ = 247 kJ mol$^{-1}$)[12–15]. This normally requires a high energy consumption and rigorous reaction temperature (>800 °C) to maintain favorable catalytic activity, but suffers from serious catalyst deactivation due to nanoparticle agglomeration and carbon deposition[16–18]. In this case, great efforts have been focused on the exploration of catalysts towards DRM reaction, such as supported noble metals (e.g., Pt[19], Ru[9,20], and Pd[21,22]) and non-noble metals (e.g., Ni[23–25] and Co[26]) catalysts. Although considerable advances have been made, rational design and preparation of highly efficient catalysts to acquire high activity and stability simultaneously, still remain a big challenge.

In general, pure metal surfaces exhibit low reactivity towards methane dissociation and are prone to deactivation resulting from carbon deposition; whilst both experimental and theoretical studies have shown that C−H bond activation is more sensitive to coordinatively unsaturated metallic sites[12,27]. In this respect, the emerging strong metal-support interaction (SMSI) has demonstrated many appealing advantages, such as the interfacial structure and synergistic

[1]State Key Laboratory of Chemical Resource Engineering, Beijing Advanced Innovation Center for Soft Matter Science and Engineering, Beijing University of Chemical Technology, 100029 Beijing, P. R. China. [2]State Key Laboratory of Heavy Oil Processing, China University of Petroleum (Beijing), 102249 Beijing, P. R. China. [3]Quzhou Institute for Innovation in Resource Chemical Engineering, 324000 Quzhou, P. R. China. [4]Shanghai Synchrotron Radiation Facility, Shanghai Institute of Applied Physics, Chinese Academy of Sciences, 201204 Shanghai, P. R. China. ✉e-mail: cui@cup.edu.cn; yangyusen@buct.edu.cn; weimin@mail.buct.edu.cn

catalysis, which have attached widespread research interest in various heterogeneous reactions (e.g., $CO_2$ methanation and water gas shift reaction)[28–31]. The fine-tuning for SMSI has been successfully employed to optimize geometric/electronic structure of metal species at the interface[32–34], which provides great opportunities to promote catalytic performance towards DRM reaction. On the one hand, the oxidic $M^{\delta+}$ metal species formed at the interfacial sites as an electron−acceptor, would reduce the $T_d$ symmetry structure of methane molecule and thus facilitate its activation dehydrogenation to $CH_x$[6,12,20]. For instance, Pirovano et al. reported that $Ni^{2+}$ species promotes C−H bond dissociation at a lower temperature relative to metal Ni based on experiments and DFT calculations[35]. On the other hand, reducible supports (e.g., $CeO_2$, $ZrO_2$, and $TiO_2$), which renders a facile conversion between two oxidation states (e.g., $Ce^{4+}$ and $Ce^{3+}$), would stabilize oxidic $M^{\delta+}$ species via accommodating metal-to-support electron transfer[32,36–38]. Meanwhile, the concomitant oxygen vacancies make a great contribution to elevate the activation adsorption of C=O group and facilitate the transformation of intermediates[34,39,40]. For example, Liu et al. reported the oxygen vacancies on $CeO_2$ surface serve as active

center towards $CO_2$ hydrogenation to methanol, where the catalytic activity is highly correlated with the oxygen vacancies concentration[41]. This evokes us to design a suitable metal-support interface structure with synergistic catalysis effect, so as to simultaneously promote catalytic activity and stability for DRM reaction and further reveal the structure-property correlation at molecular/atomic scale.

Herein, we report an Ir nanoclusters supported on $CeO_2$ catalyst prepared through a facile impregnation-reduction method. HAADF-STEM, *quasi* in situ XPS and in situ XAFS confirm the formation of interface structure ($Ir^{\delta+}−O_v−Ce^{3+}$), whose concentration can be modulated via adjusting the Ir loading. The optimal catalyst 0.6% Ir/$CeO_{2−x}$ (Fig. 1a) exhibits high conversions of $CH_4$ (~72%) and $CO_2$ (~82%) at 700 °C, with a $CH_4$ reaction rate of ~973 $\mu mol_{CH4} \, g_{cat}^{-1} \, s^{-1}$; and a 100 h stream-on-line test demonstrates a satisfactory stability without obvious deactivation. This is, to the best of our knowledge, preponderant to the state-of-the-art catalysts under similar reaction conditions. Kinetics studies verify that the dissociation of $CH_4$ is the rate-determining step in DRM reaction, whose activation energy decreases significantly by ~50 kJ $mol^{-1}$ owing to the interfacial

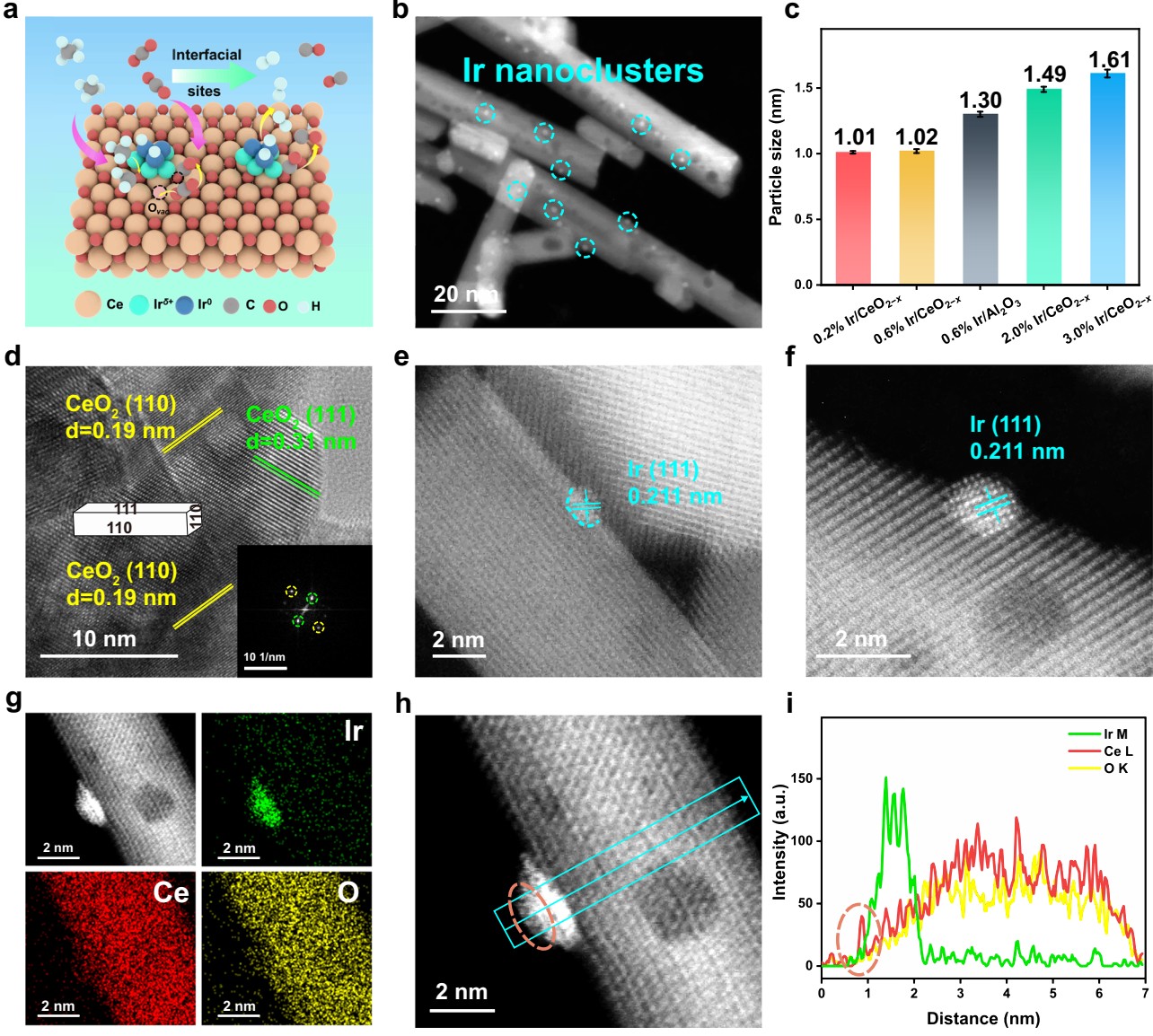

**Fig. 1 | Microstructure and morphology studies of Ir/CeO₂₋ₓ samples.**
**a** Schematic illustration of Ir/$CeO_{2−x}$ samples. **b**, **d** TEM and HR-TEM images of 0.6% Ir/$CeO_{2−x}$. **c** Particle size of various Ir/$CeO_{2−x}$ and Ir/$Al_2O_3$ samples determined by TEM. **e**, **f** High-resolution AC-HAADF-STEM images of 0.2% and 0.6% Ir/$CeO_{2−x}$, respectively. **g**, **h** AC-HAADF-STEM image and corresponding EDS mapping of 0.6% Ir/$CeO_{2−x}$. **i** Corresponding elemental line scanning of 0.6% Ir/$CeO_{2−x}$.

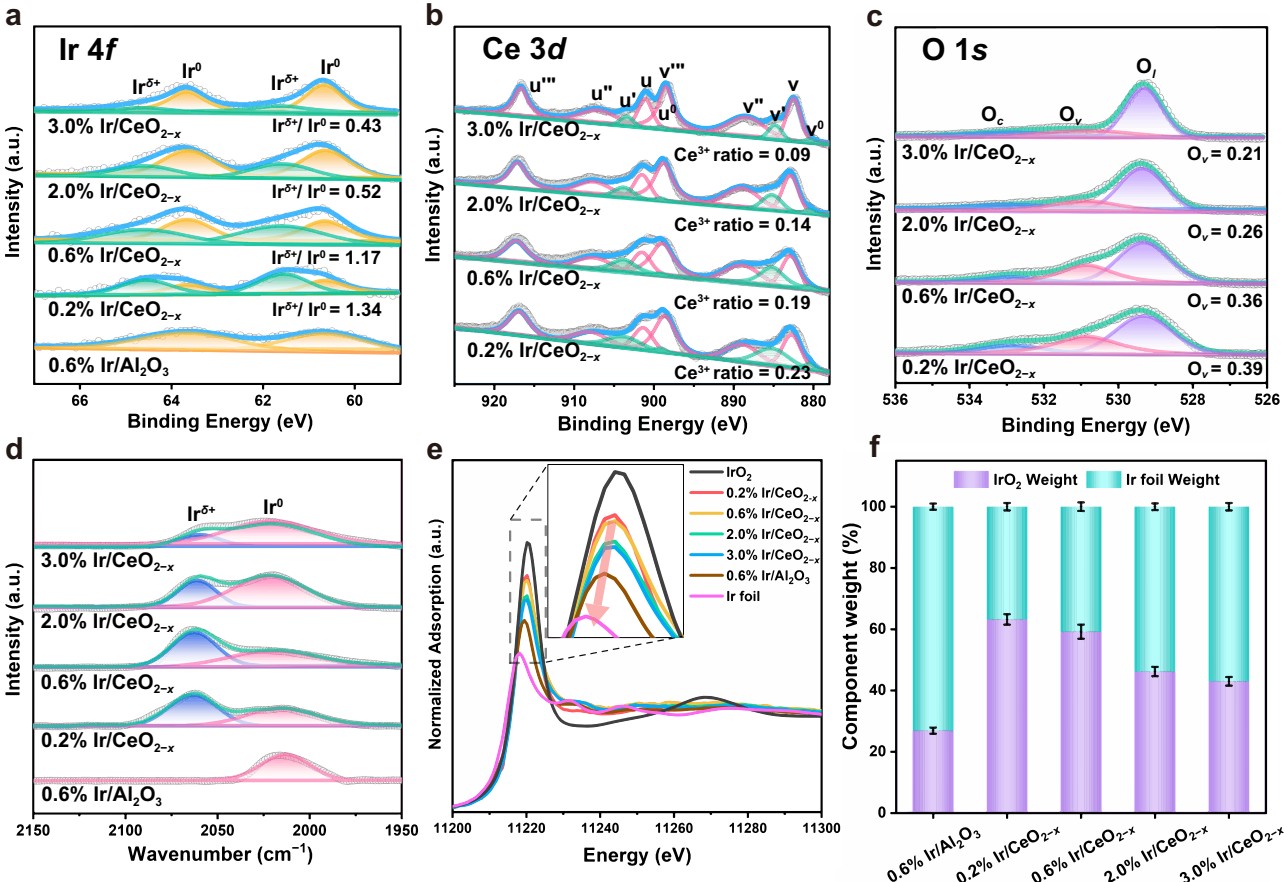

**Fig. 2 | Fine-structure characterizations of Ir/Al₂O₃ and Ir/CeO₂₋ₓ samples.** **a–c** *Quasi* in situ XPS of Ir 4*f*, Ce 3*d* an O 1*s* for Ir/Al₂O₃ and Ir/CeO₂₋ₓ samples with various Ir loading. **d** In situ CO-DRIFTS spectra on the surface over Ir/Al₂O₃ and various Ir/CeO₂₋ₓ samples. **e** Ir L₃-edge XANES spectra and (**f**) diagram of the linear combination fitting (LCF) results for various samples.

synergistic catalysis. *Operando* investigations (DRIFTS and XAFS), catalytic evaluations and DFT calculations substantiate that the interfacial sites (Ir$^{\delta+}$–O$_v$–Ce$^{3+}$) serve as the intrinsic active center: CH₄ molecule undergoes activation adsorption and dissociation to CH₂* species and H₂ at the interfacial Ir$^{\delta+}$ site, and then CH₂* experiences oxidation by neighboring oxygen species to generate CH₂O*, followed by CH₂O* dehydrogenation to produce CO and H₂; the concomitant O$_v$ is replenished by the activation adsorption of C=O group in CO₂. This interfacial synergistic catalysis not only enhances the catalytic activity for DRM reaction, but also inhibits catalyst deactivation from excessive decomposition of CH₂* species to carbon deposition.

## Results and discussion
### Structural characterizations of catalysts
Both Ir/Al₂O₃ and Ir/CeO₂₋ₓ samples with various Ir loading (0.2%–3%) were prepared via a facile impregnation-reduction method, whose XRD patterns (Supplementary Fig. 1) displayed a series of characteristic reflections indexed to a typical Al₂O₃ (JCPDS 77-0396) and CeO₂ (JCPDS 78-0694) phase, respectively. No recognizable diffractions peaks of Ir or IrO₂ are found for these samples, implying a highly dispersed Ir species and/or a low Ir content. TEM images of Ir/CeO₂₋ₓ samples (Fig. 1b) show numerous Ir nanoclusters are well dispersed and anchored onto the CeO₂ nanorods support, in which the mean particle size of Ir increases from ~1.0 nm (0.2% and 0.6% Ir/CeO₂₋ₓ) to ~1.5 nm (2% Ir/CeO₂₋ₓ) and then to ~1.6 nm (3.0% Ir/CeO₂₋ₓ) (Fig. 1c and Supplementary Fig. 4−7). Accordingly, the dispersion of Ir ($D_{Ir}$, Supplementary Table 1) decreases gradually from 82% (0.2% Ir/CeO₂₋ₓ) to 39% (3.0% Ir/CeO₂₋ₓ). From the local magnification HR-TEM images (Fig. 1d and Supplementary Fig. 3−7), two clear crystalline phases are

identified as ~0.19 and ~0.31 nm, respectively, corresponding to the (110) and (111) planes of CeO₂ nanorods support. As shown in Supplementary Fig. 8, the (110) facet is predominantly exposed accompanied with minor (111) facet, consistent with the previous reports[41,42]. As a control sample, 0.6% Ir/Al₂O₃ displays a larger particle size (~1.3 nm) and a lower dispersion (~65%) relative to 0.6% Ir/CeO₂₋ₓ, along with lattice spacings of ~0.19 and ~0.23 nm ascribed to (400) and (311) planes of Al₂O₃[43]. In addition, the aberration-correction high-angle annular dark-field scanning transmission electron microscopy (AC-HAADF-STEM) was conducted to explore detailed structure of Ir/CeO₂₋ₓ. As shown in Fig. 1e and f, a clear lattice fringe (~0.211 nm) indexed to Ir(111) plane is observed on the surface of CeO₂ for both the 0.2% and 0.6% Ir/CeO₂₋ₓ samples. Moreover, the energy dispersive spectroscopy (EDS) elemental mapping and elemental line scanning of 0.6% Ir/CeO₂₋ₓ sample (Fig. 1g−i) show a partial coating of CeO₂ on the surface of Ir cluster, indicating the formation of interfacial structure between Ir and CeO₂.

*Quasi* in situ XPS spectra were performed to investigate the electronic structure of surface Ir species. As shown in Fig. 2a, the Ir/Al₂O₃ sample displays two peaks at 60.6 eV (Ir 4*f*₇/₂) and 63.6 eV (Ir 4*f*₅/₂) corresponding to the Ir⁰ species. In contrast, for the four Ir/CeO₂₋ₓ samples, besides the same Ir⁰ peaks, two additional strong peaks at 61.6 eV (Ir 4*f*₇/₂) and 64.6 eV (Ir 4*f*₅/₂) are found, which are attributed to the Ir$^{\delta+}$ species[44–46]. This indicates the electron transfer from Ir species to CeO₂ support at the interface via the electronic metal-support interaction (EMSI), which is absent in the Ir/Al₂O₃ sample. With the increase of Ir content, the ratio of Ir$^{\delta+}$/(Ir$^{\delta+}$+Ir⁰) declines gradually from 57% (0.2% Ir/CeO₂₋ₓ) to 30% (3.0% Ir/CeO₂₋ₓ) (Supplementary Table 2), as a result of the decreased Ir dispersion degree. Furthermore, in situ

CO-DRIFTS is implemented to investigate the configuration of Ir species (Fig. 2d), from which a broad band centered at -2020 cm$^{-1}$ due to the linear CO at Ir$^0$ site is found for the Ir/Al$_2$O$_3$ sample. Notably, in the case of Ir/CeO$_{2-x}$ samples, both the linear adsorption of CO at Ir$^0$ (-2020 cm$^{-1}$) and gem-dicarbonyl species adsorption at Ir$^{\delta+}$ (-2060 cm$^{-1}$) are observed[44,45,47]. With the increment of Ir loading, according to the Gaussian peak fitting results, the relative peak intensity of Ir$^{\delta+}$/(Ir$^{\delta+}$+Ir$^0$) displays an obvious decrease from 0.2% Ir/CeO$_{2-x}$ (56%) to 3.0% Ir/CeO$_{2-x}$ (23%) (Supplementary Table 3), in accordance with the variation tendency of Ir$^{\delta+}$/(Ir$^{\delta+}$+Ir$^0$) in the XPS results.

X-ray absorption near-edge structure (XANES) measurements at normalized Ir L$_3$-edge were implemented to analyze the electronic state and coordination fine structure. As shown in Fig. 2e, the white line peaks of Ir/CeO$_{2-x}$ and Ir/Al$_2$O$_3$ samples are located between Ir foil and IrO$_2$ reference, suggesting the existence of positively charged Ir species. Moreover, the intensity of white line declines gradually from 0.2% Ir/CeO$_{2-x}$ to 3.0% Ir/CeO$_{2-x}$ and then to Ir/Al$_2$O$_3$, indicating the decrease in the oxidation state of Ir$^{\delta+}$ species (reduced interfacial electron transfer) along with the increase in metallic Ir$^0$. The Fourier transforms of the extended X-ray absorption fine spectra (EXAFS) in the R space (Supplementary Fig. 10) show that all these samples exhibit coexistent of Ir–O scattering (-1.5 Å) and Ir–Ir scattering (-2.5 Å). Accordingly, we conducted the linear combination fitting (LCF) of XANES (Fig. 2f) to determine the Ir species composition in these samples. The control sample Ir/Al$_2$O$_3$ displays a low Ir$^{4+}$ atomic ratio of 27%. In contrast, the Ir$^{4+}$ is predominant for the Ir/CeO$_{2-x}$ samples, in which the Ir$^{4+}$ atomic ratio of 0.2% and 0.6% Ir/CeO$_{2-x}$ samples are 63% and 59% (Supplementary Table 4), respectively; whilst the Ir$^0$ plays a leading role for the 2.0% and 3.0% Ir/CeO$_{2-x}$ samples, as a result of the increased particle size of Ir. The average oxidation state of iridium species is calculated based on the results from LCF analysis, which gives the following sequence: 0.2% Ir/CeO$_{2-x}$ (+2.5) > 0.6% Ir/CeO$_{2-x}$ (+2.4) > 2.0% Ir/CeO$_{2-x}$ (+1.8) > 3.0% Ir/CeO$_{2-x}$ (+1.7) > 0.6% Ir/Al$_2$O$_3$ (+1.1).

Then, we used *quasi* in situ XPS to study the defective sites of Ir/CeO$_{2-x}$ samples, which contribute to mediate the CO$_2$ activation. For the pristine CeO$_2$ support (Supplementary Fig. 13), the spectra of Ce 3$d$ show six strong peaks at 882.9, 889.0, 898.6 eV (marked as v, v″, and v‴, respectively) and 901.5, 907.6, 917.2 eV (marked as u, u″, and u‴, respectively) assigned to 3$d^{10}$4$f^0$ state of Ce$^{4+}$ species, with a spin-orbit splitting of about 18.6 eV. In terms of Ir/CeO$_{2-x}$ samples (Fig. 2b), four peaks appear at 880.5 eV, 885.2 eV, 899.1 eV, and 903.8 eV (marked as v$^0$, v′, u$^0$, and u′, respectively) belonging to 3$d^{10}$4$f^1$ state of Ce$^{3+}$[41,48]. The relative concentration of surface Ce$^{3+}$, calculated by Ce$^{3+}$/(Ce$^{3+}$ + Ce$^{4+}$) based on corresponding peak areas (Supplementary Table 2), decreases gradually from 23% (0.2% Ir/CeO$_{2-x}$) to 10% (3.0% Ir/CeO$_{2-x}$). Moreover, the oxygen vacancies of Ir/CeO$_{2-x}$ and Ir/Al$_2$O$_3$ samples were further analyzed via deconvolution of *quasi* in situ O 1$s$ XPS spectra. As shown in Fig. 2c, three peaks are found at 529.3, 530.8 and above 532.8 eV, which are assigned to the lattice oxygen (O$_l$), oxygen vacancies (O$_v$) and other weakly bound oxygen species (O$_c$, such as hydroxyl oxygen or chemisorbed oxygen species), respectively[48,49]. The relative ratio of oxygen vacancy (O$_v$) calculated as O$_v$/(O$_l$ + O$_v$ + O$_c$) ranks in the following order: 0.2% Ir/CeO$_{2-x}$ (39%) > 0.6% Ir/CeO$_{2-x}$ (36%) > 2.0% Ir/CeO$_{2-x}$ (26%) > 3.0% Ir/CeO$_{2-x}$ (21%) (Supplementary Table 2), in agreement with the tendency of Ce$^{3+}$/(Ce$^{3+}$ + Ce$^{4+}$) ratio. Therefore, in situ CO-DRIFTS, XAFS and *quasi* in situ XPS results notarize the formation of interface structure (Ir$^{\delta+}$–O$_v$–Ce$^{3+}$) originating from SMSI, whose relative concentration decreases with increment of Ir loading.

## Catalytic evaluations

The preceding samples were evaluated for DRM under a gas hourly space velocity as high as 240000 mL g$^{-1}$ h$^{-1}$ at atmospheric pressure. As shown in Fig. 3a and b, both the CH$_4$ and CO$_2$ conversions as a function of reaction temperature show a positive correlation for these samples, duo to the strong endothermic characteristic. The control sample 0.6% Ir/Al$_2$O$_3$ gives a normal catalytic performance towards DRM reaction; whilst the catalytic performance of Ir/CeO$_{2-x}$ samples improve significantly. Notably, the catalytic activity exhibits a volcanic curve at each reaction temperature along with the increase of Ir loading: an increase from 0.2% Ir/CeO$_{2-x}$ to 0.6% Ir/CeO$_{2-x}$ (the maximum value) is present, followed by a slight descend to 2.0% Ir/CeO$_{2-x}$ and 3.0% Ir/CeO$_{2-x}$. As for the optimal 0.6% Ir/CeO$_{2-x}$ sample, both the CH$_4$ and CO$_2$ conversions reach up to the thermodynamic equilibrium, and the reaction rate is 3−20 times higher than previously reported studies under similar reaction conditions within 650−750 °C (Fig. 3c and Supplementary Table 5)[19,50]. Specifically, the 0.6% Ir/CeO$_{2-x}$ catalyst exhibits high conversions of CH$_4$ (72%) and CO$_2$ (82%) with a CH$_4$ reaction rate of -973 μmol$_{CH4}$ g$_{cat}^{-1}$ s$^{-1}$ at a relatively moderate temperature (700 °C), which are precedent to the state-of-the-art catalysts[6,7,9,16,20,24,25,49−51]. In addition, the long-term stability test displays a rapid deactivation for the Ir/Al$_2$O$_3$ sample within 15 h due to Ir agglomeration and carbon deposition at 700 °C (Supplementary Fig. 16 and 17). In contrast, both the CH$_4$ and CO$_2$ conversions of 0.6% Ir/CeO$_{2-x}$ catalyst remain almost unchanged within 100 h on stream (Fig. 3d). Moreover, the used 0.6% Ir/CeO$_{2-x}$ catalyst does not show obvious structural change compared with the fresh sample, verified by TEM, XPS and in situ CO-DRIFTS (Supplementary Fig. 18−20), indicating a satisfactory stability in DRM reaction. The results above demonstrate excellent performance of 0.6% Ir/CeO$_{2-x}$ catalyst, which shows potential application in industrial applications.

Furthermore, we performed kinetic studies on CH$_4$ and CO$_2$ activation as well as the rate-determining step in DRM system. Firstly, the effects of external and internal diffusion limitation have been eliminated under the aforementioned reaction conditions[34,50,52]. On this basis, the kinetic experimental data were studied via setting a stationary partial pressure of one reactant whilst changing the other partial pressure (Fig. 3e, f), and the obtained results were calculated for kinetic parameters and were shown in Supplementary Table 6. The reaction rate over 0.6% Ir/CeO$_{2-x}$ catalyst displays a linear positive correlation with the partial pressure of CH$_4$ and CO$_2$. Nevertheless, the calculated reaction order with respect to CH$_4$ (-0.67 and -0.53) is significantly higher than that of CO$_2$ (-0.09 and -0.07), indicating that the CH$_4$ activation is critical to the reaction kinetics, consistent with previous studies[50,52,53]. Moreover, the apparent activation energy ($E_a$) of CH$_4$ over 0.6% Ir/CeO$_{2-x}$ is 91 kJ mol$^{-1}$, much larger than that of CO$_2$ (70 kJ mol$^{-1}$) (Fig. 3g and Supplementary Fig. 22 and 23). The results verify that the CH$_4$ dissociation entails a higher energy barrier and serves as the rate-determining step in this catalytic system. Notably, the $E_a$ value on 0.6% Ir/CeO$_{2-x}$ catalyst shows a marked decline by 35% relative to the 0.6% Ir/Al$_2$O$_3$ sample, which indicates the interfacial sites play a critical role in activating reactant molecule. In addition, the intrinsic TOF of CH$_4$ is evaluated at a low conversion (below 15%, Fig. 3h), which gives a decrease order as follows: 0.2% Ir/CeO$_{2-x}$ (168 mol$_{CH4}$ mol$_{Ir}^{-1}$ s$^{-1}$) > 0.6% Ir/CeO$_{2-x}$ (163 mol$_{CH4}$ mol$_{Ir}^{-1}$ s$^{-1}$) > 2.0% Ir/CeO$_{2-x}$ (122 mol$_{CH4}$ mol$_{Ir}^{-1}$ s$^{-1}$) > 3.0% Ir/CeO$_{2-x}$ (110 mol$_{CH4}$ mol$_{Ir}^{-1}$ s$^{-1}$). To further the reveal correlation of intrinsic active site and structure-property, the intrinsic TOF of CH$_4$ is plotted as a function of interfacial Ir$^{\delta+}$ concentration (based on the results of in situ CO-DRIFTS), from which an approximative linear relationship is present (Fig. 3i). Furthermore, a positive correlation between intrinsic TOF and surface Ce$^{3+}$ ratio (Supplementary Fig. 24) or surface oxygen vacancy ratio (Supplementary Fig. 25) is also demonstrated. The results above elucidate that the Ir$^{\delta+}$–O$_v$–Ce$^{3+}$ interfacial sites serve as the intrinsic active center towards DRM reaction, accounting for the prominent catalytic performance.

**Catalytic mechanism.** In situ/*operando* XANES of Ir L$_3$-edge and Ce L$_3$-edge combined with *quasi* in situ XPS were applied to reveal the

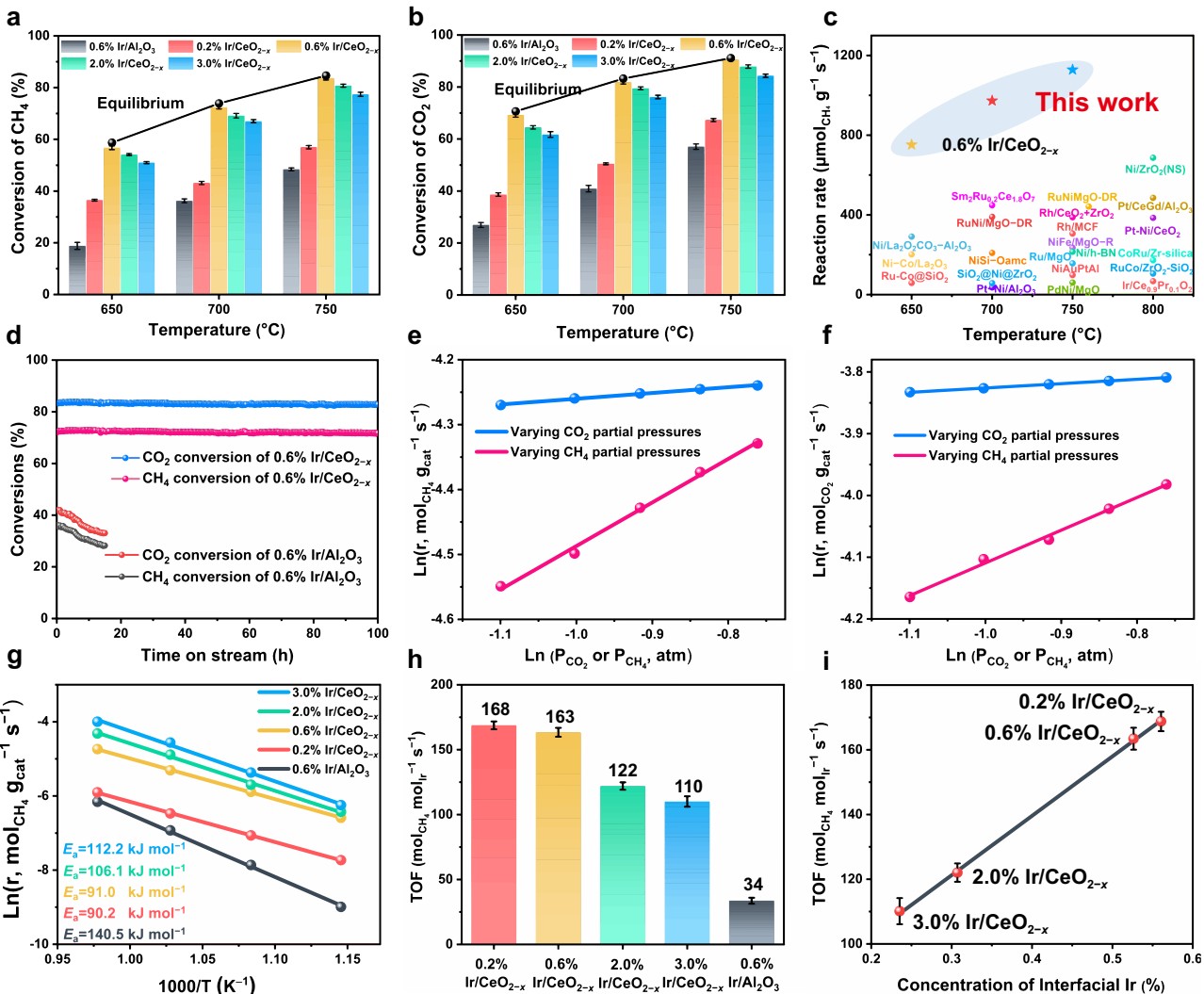

**Fig. 3 | Catalytic performance of various catalysts. a, b** CH$_4$ conversion and CO$_2$ conversion over various samples at 650, 700, and 750 °C. **c** Comparison study on mass specific activity between 0.6% Ir/CeO$_{2-x}$ and other typical catalysts used in DRM reaction[64–79]. **d** Stability test of 0.6% Ir/CeO$_{2-x}$ and 0.6% Ir/Al$_2$O$_3$ at 700 °C for DRM. Evaluated conditions: CH$_4$/CO$_2$/N$_2$ = 20/20/5 mL min$^{-1}$, GHSV = 240000 mL g$^{-1}$ h$^{-1}$. **e, f** Correlation of CH$_4$ or CO$_2$ partial pressure on the reaction rates of CH$_4$ and CO$_2$. **g** Kinetic studies and calculated activation energy ($E_a$) of CH$_4$ over various catalysts. **h** Intrinsic TOF over various catalysts within the catalytic dynamic range. **i** TOF as a function of interfacial Ir concentration calculated by CO-DRIFTS results.

dynamic variation of fine structure and electronic interaction at interfacial sites under the catalytic reaction. During the measurement, CH$_4$ and CO$_2$ was introduced into the reaction cell in turn, and the catalytic reaction was triggered at 700 °C via injecting the second reactant gas, so as to observe the formation and variation of interface structure (Ir$^{\delta+}$–O$_\nu$–Ce$^{3+}$). When CH$_4$ is introduced alone, the white line peaks of Ir and Ce shift close to the reference Ir foil and CeF$_3$ (Fig. 4a, d), respectively, indicating a decline in valence states of Ir and Ce (Fig. 4b, e). The corresponding variations in XPS spectra of Ir 4$f$ and Ce 3$d$ are also observed (Fig. 4c, f): the Ir$^{\delta+}$/(Ir$^{\delta+}$+Ir$^0$) ratio decreases whilst the Ce$^{3+}$/(Ce$^{3+}$ + Ce$^{4+}$) ratio and O$_\nu$ increase. This implies the occurrence of CH$_4$ dissociation to CH$_x{}^*$ species, which then combines with surface reactive O to generate more Ir$^{\delta+}$–O$_\nu$–Ce$^{3+}$ interface sites. After the injection of CO$_2$, the white line peaks of Ir and Ce shift back to their original position, indicating the replenishment of O$_\nu$ by CO$_2$. In Fig. 4a, d, as CO$_2$ is introduced alone, the white line peaks of Ir and Ce move close to the reference IrO$_2$ and CeO$_2$, respectively (elimination of primary O$_\nu$); and XPS results show the increased valence states of Ir and Ce accompanied with reduced Ce$^{3+}$/(Ce$^{3+}$ + Ce$^{4+}$) ratio and O$_\nu$ (Fig. 4b, e). Afterwards, the subsequent CH$_4$ flowing induces the recovery of Ir and Ce white line peaks to their original position,

corresponding to the CH$_4$ dissociation assisted with surface reactive oxygen species.

In situ/*operando* DRIFTS experiments of reactants were carried out to further identify the intermediate species and monitor the evolution of dynamic reaction process at the interface structure Ir$^{\delta+}$–O$_\nu$–Ce$^{3+}$ (Fig. 5a–h). When CH$_4$ is introduced individually into the reactor at 700 °C, in addition to the gas phase CH$_4$ at ~3016 and ~1304 cm$^{-1}$, another two bands at ~1330 and ~1350 cm$^{-1}$ corresponding to the deformation vibration of CH$_2{}^*$ and CH$_3{}^*$ are observed[4,20,51], respectively, due to the activation adsorption and dissociation of CH$_4$ at interface Ir$^{\delta+}$ sites. Subsequently, with the injection of CO$_2$, two strong peaks located at ~2360 and ~1550 cm$^{-1}$, as well as another broad one at ~3750−3550 cm$^{-1}$ appear, which are attributed to the gas phase CO$_2$, the monodentate carbonate species (HCOO*) and surface hydroxyl group (OH*), respectively (Fig. 5b–d and Supplementary Figs. 28, 29)[16,20,54,55]. Notably, another IR band assigned to the CH$_x$O* species is found at ~1390 cm$^{-1}$, accompanied with the weakened bands of CH$_4$ and CH$_x{}^*$ species[56,57]. This is probably due to the oxidation of CH$_x{}^*$ by reactive oxygen species originating from CO$_2$ dissociation. In addition, three bands between ~2200 and ~2000 cm$^{-1}$ are detected, which are ascribed to gaseous CO and adsorbed CO* at

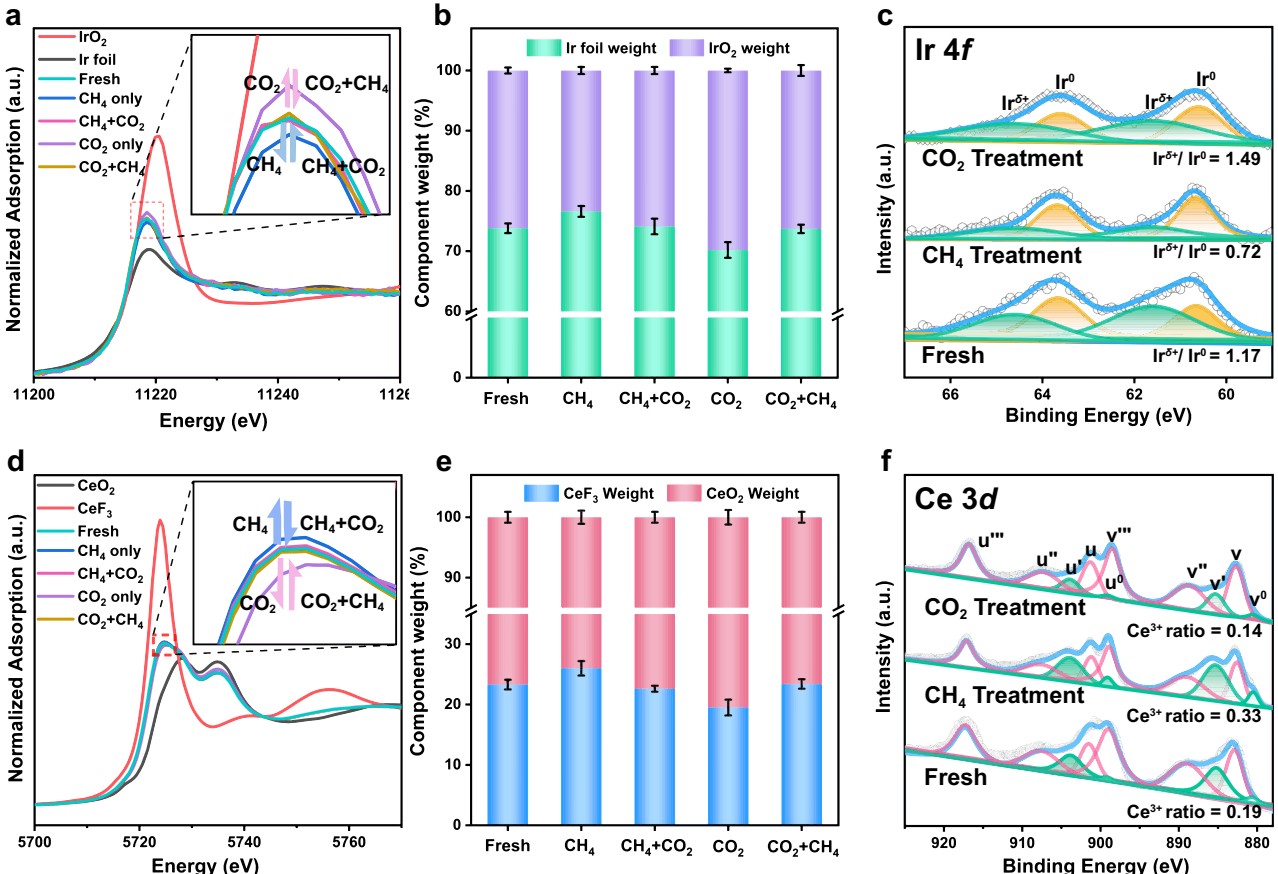

**Fig. 4 | Local coordination structure and surface structure of 0.6% Ir/CeO$_2$ during DRM reaction. a, b** In situ/*operando* normalized XANES at Ir L$_3$-edge and diagram of the linear combination fitting (LCF) results of 0.6% Ir/CeO$_{2-x}$ with CH$_4$, CO$_2$ and CH$_4$ + CO$_2$ treatment, respectively. **d, e** Ce L$_3$-edge and diagram of the linear combination fitting (LCF) results of 0.6% Ir/CeO$_{2-x}$ with CH$_4$, CO$_2$ and CH$_4$ + CO$_2$ treatment, respectively. **c, f** *Quasi* in situ XPS spectra of Ir 4$f$ and Ce 3$d$ for the fresh 0.6% Ir/CeO$_{2-x}$ and the same catalyst after CH$_4$ or CO$_2$ treatment at 700 °C.

Ir$^{\delta+}$, respectively (Fig. 5c)[44–46]. Once the atmosphere is switched from the mixture gas (CO$_2$ and CH$_4$) to individual CH$_4$, the bands of gas phase CO$_2$ weaken firstly, and then the bands assigned to CH$_x$O*, HCOO*, OH* and CO species disappear gradually accompanied with the enhancement of CH$_4$ and CH$_x$* peaks. This demonstrates the oxygen-containing species (CH$_x$O*, HCOO*, OH*) serves as important intermediate, whose consumption can be reproduced by CO$_2$ at interface O$_\nu$.

Next, we changed the study paradigm, in which CO$_2$ is injected into the reactor firstly under the same conditions. Accordingly, the bands assigned to CO$_2$ is observed (Supplementary Fig. 30 and Fig. 5e). With the subsequent flowing of CH$_4$, the bands of CH$_4$, CH$_x$* and CH$_3$* species are found (Fig. 5g, h), followed by the emergence of CH$_x$O* and CO peaks as well as the weakened OH* band. This verifies the significance of CH$_x$O* species originating from the reaction between CH$_x$* and surface oxygen species, in accordance with the results of Fig. 5a–d. *Operando* investigations above (XAFS and DRIFTS) substantiate that the interface structure (Ir$^{\delta+}$–O$_\nu$–Ce$^{3+}$) serves as the intrinsic active center with a crucial synergistic effect: Ir$^{\delta+}$ promotes the activation adsorption of CH$_4$ molecule whilst CO$_2$ dissociation occurs at the Ce$^{3+}$–O$_\nu$ site, followed by the formation of the key intermediate (CH$_x$O* species).

To in-depth explore the decisive role of Ir$^{\delta+}$–O$_\nu$–Ce$^{3+}$ interfacial sites in the reaction process, DFT calculations were investigated on Ir$_7$/CeO$_{2-x}$ model based on the experimental results (Supplementary Fig. 31). As shown in Fig. 5i and Supplementary Fig. 32, firstly, CH$_4$ molecule undergoes adsorption at the interfacial Ir$^{\delta+}$ of Ir$_7$/CeO$_{2-x}$ (110) with a small adsorption energy (−0.03 eV); then, the C–H bond cleavage of CH$_4$ occurs to generate CH$_3$* (TS1: 1.12 eV). Afterwards, the

CH$_3$* species experiences dehydrogenation process which shows an energy barrier of 1.43 eV, excluding the oxidation of CH$_3$* to CH$_3$O* with a large steric hindrance. Subsequently, two possible steps are involved: (1) CH$_2$* oxidation to CH$_2$O* and (2) CH$_2$* dehydrogenation to CH*. However, the former displays a much lower energy barrier (TS3: 1.03 eV) than the latter (TS4: 1.56 eV), in agreement with the formation of CH$_x$O* intermediate verified by the *operando* DRIFTS results. This step is crucial, which inhibits excessive decomposition of CH$_2$* species to carbon deposition. The next dehydrogenation of CH$_2$O* to CHO* (TS5) and CO (TS6) shows normal activation barriers of 0.63 and 0.73 eV, respectively. Finally, the produced CO undergoes desorption from the O$_\nu$ and the remaining four active hydrogen form into two H$_2$ molecules (Supplementary Fig. 32). Meanwhile, CO$_2$ molecule experiences dissociation adsorption at the O$_\nu$ on the surface with an adsorption energy of −1.85 eV and an energy barrier of 0.7 eV (Supplementary Fig. 33 and 34), with the formation of active oxygen species that participates in the CH$_2$* oxidation to CH$_2$O*. According to the calculation results, the dehydrogenation of CH$_3$* species to CH$_2$* gives the highest energy barrier (1.43 eV), which is determined as the rate-determining step of DRM reaction, in accordance with the experimental results (Fig. 3e–f). In addition, a comparative study between Ir$_7$/CeO$_{2-x}$ and Ir$_7$/Al$_2$O$_3$ shows that the reaction energy barrier of rate-determining step in the former case (1.43 eV) is significantly lower than the latter one (Supplementary Figs. 35, 36: 2.88 eV), demonstrating the essential contributions of interface sites (Ir$^{\delta+}$–O$_\nu$–Ce$^{3+}$), in well agreement with the catalytic performance in Fig. 3a–h.

In summary, we report an Ir/CeO$_{2-x}$ catalytic system with metal–support interface structure towards DRM reaction. The

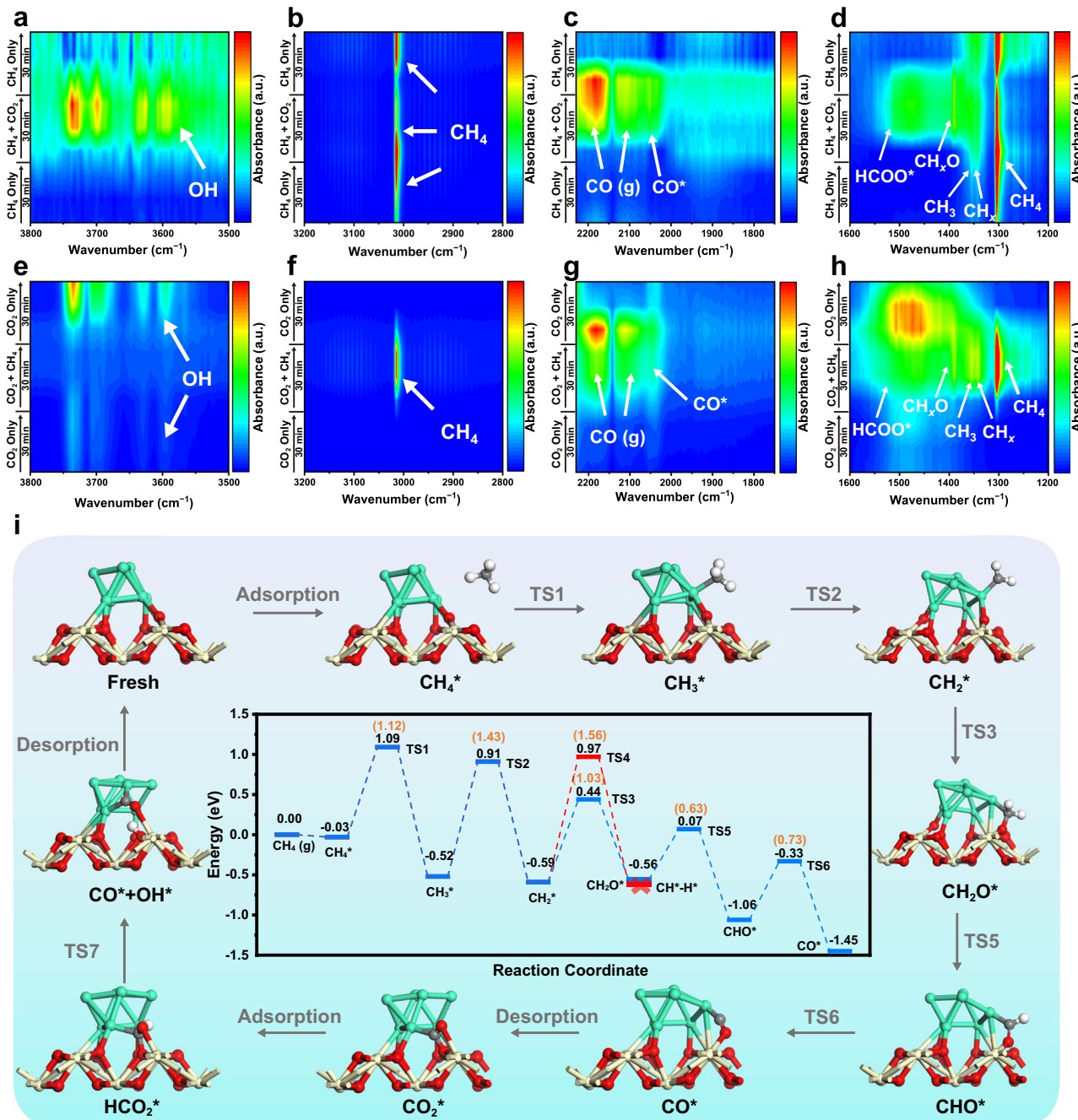

**Fig. 5 | In situ/operando DRIFTS spectra and DFT calculations of DRM reaction on 0.6% Ir/CeO$_{2-x}$.** In situ/operando DRIFTS spectra over 0.6% Ir/CeO$_{2-x}$ at 700 °C after in-situ pretreatment and He purging, followed by exposure to: (**a–d**) first CH$_4$ atmosphere, subsequent CH$_4$ + CO$_2$ and then CH$_4$ atmosphere for 30 min, respectively; (**e–h**) first CO$_2$ atmosphere, subsequent CO$_2$ + CH$_4$ and then CO$_2$ atmosphere for 30 min, respectively. **i** Schematic illustration for DRM reaction at the interface of Ir/CeO$_{2-x}$. Ir, green; Ce, yellow; C, gray; O, crimson; H, white. The inset shows potential energy profile of CH$_4$ decomposition by Ir/CeO$_{2-x}$(110). 'TS' represents a transition state. The black and orange numbers represent the adsorption energies and energy barriers of the elementary steps, respectively.

obtained 0.6% Ir/CeO$_{2-x}$ catalyst exhibits exceptional conversions of CH$_4$ (72%) and CO$_2$ (82%), a CH$_4$ reaction rate of ~973 μmol$_{CH4}$ g$_{cat}^{-1}$ s$^{-1}$ and a satisfactory service stability within 100 h at a relatively low temperature (700 °C). A joint investigation based on HAADF-STEM, *quasi* in situ XPS and in situ XAFS confirms the formation of interface structure (Ir$^{\delta+}$−O$_\upsilon$−Ce$^{3+}$), whose concentration can be modulated via adjusting the Ir loading. *Operando* investigations (DRIFTS and XAFS), catalytic evaluations and DFT calculations substantiate that the interfacial sites (Ir$^{\delta+}$−O$_\upsilon$−Ce$^{3+}$) serve as the intrinsic active center to facilitate the dissociation of CH$_4$ (the rate-determining step) and the oxidation of CH$_2$* to CH$_2$O*; the concomitant O$_\upsilon$ can be replenished by the activation

adsorption of CO$_2$. This interfacial synergistic catalysis plays a crucial role in boosting the catalytic performance and inhibiting deactivation, which paves a way for the design of other high-performance heterogeneous catalysts towards structure-sensitive reactions.

## Methods
### Chemicals and materials
Analytical grade chemical reagents were purchased in Aladdin company and used directly without further purification, including: Ce(NO$_3$)$_3$·6H$_2$O, NaOH, Al$_2$O$_3$, and H$_2$IrCl$_6$·6H$_2$O. Deionized water was adopted in all experiment steps.

## Preparation of catalysts

$CeO_2$ nanorods were prepared via a hydrothermal method reported by our group[42]. Typically, $Ce(NO_3)_3$ solution (0.4 M, 20 mL) and NaOH solution (6.8 M, 140 mL) were fully mixed with vigorous stirring for 30 min at room temperature. The obtained milky slurry was placed into a 200 mL sealed Teflon autoclave for 24 h at 100 °C. After filtering, washing thoroughly, and drying at 65 °C for 18 h, the sample was calcined in air at 500 °C with a heating rate of 10 °C min$^{-1}$ for 4 h to obtain the $CeO_2$ nanorods support. $CeO_2$ (0.5 g) was dispersed into deionized water (20 ml) and $H_2IrCl_6 \cdot 6H_2O$ aqueous solution (0.022 g mL$^{-1}$; 0.105, 0.315, 1.050, 1.575 mL, respectively) was slowly dripped into above solution with vigorous stirring for various Ir loading samples. After 8 h of reaction, the resulting precipitate was centrifuged, washed thoroughly with deionized water and ethanol, followed by drying at 60 °C for 12 h. Before the DRM reaction, the sample was pre-treated at 750 °C for 3 h in a gaseous mixture of $CH_4$ and $CO_2$ (1:1, v/v; total flow rate: 50 mL min$^{-1}$). As a reference, the Ir/$Al_2O_3$ sample was prepared via the same method described above by using $Al_2O_3$ as the support, in which the pre-treated steps are in accordance with those of Ir/$CeO_{2-x}$ samples.

## Characterizations

X-ray diffraction (XRD) experiments were carried out with Bruker D8 Advance diffractometer. The elemental content was determined by Shimadzu ICPS-7500 equipment. The morphology and structure of catalysts were studied on JEOL JEM-2010 high-resolution transmission electron microscope. AC-HAADF-STEM images and EDS mapping data were performed on JEOL JEM-ARM200F. The CO pulses chemisorption experiments were conducted on Micromeritics Autochem II 2920. *Quasi* in situ XPS measurements were recorded on Kratos Axis Ultra DLD Instrument. The pre-treated sample was placed in a glove box and transferred into a sample rod in a $N_2$ atmosphere. In situ*/Operando* XAFS at Ir L$_3$-edge and Ce L$_3$-edge measurements were recorded at the beamline BL11B of the Shanghai Synchrotron Radiation Facility (SSRF), Shanghai Institute of Applied Physics, Chinese Academy of Sciences (CAS). In situ*/operando* DRIFTS were studied on a Bruker TENSOR II infrared spectrometer with a MCT detector. The detailed experimental methods are present in the Supplementary Information.

## DFT calculations

The density functional theory (DFT) calculations based on first-principle methodology were investigated using the Vienna ab initio simulation package (VASP 5.4.4)[58,59]. Generalized gradient approximation (GGA) of PBE functional was applied to describe the exchange and correlation energy. Grimme's DFT-D3 method and projector augmented wave (PAW) method were employed to illustrate the effect of van der Waals interaction and to depict the core electrons, respectively[60,61]. The climbing image nudged elastic band (CI-NEB) method was employed to determine reaction transition states[62,63].

# Data availability

The primary data that support the plots within this paper and other finding of this study are available from the corresponding author on reasonable request. Source data are provided with this paper.

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

## Acknowledgements

This work was supported by the National Key R&D Program of China (2021YFC2103500), the National Natural Science Foundation of China (22172006, 22102006, 22288102, and 22109177), and the Young Elite Scientists Sponsorship Program by CAST (2023QNRC001). The authors are thankful for the support of the SSRF (Shanghai Synchrotron Radiation Facility) during the XAFS measurements at the beamline of BL11B.

## Author contributions

M.W., Y.Y., G.C. and H.W. conceived the idea and designed the research. H.Y. and H.L. performed the DFT calculations. H.W., Z.L. and H.M. synthesized the catalysts and conducted the characterizations and reaction tests. L.W. and J.L. helped the STEM and XAFS analysis. All the authors analyzed the data and wrote the paper.

## Competing interests

The authors declare no competing interests.
