## [Peer Review File · Nature Communications]

Interfacial synergistic catalysis at Ir@CeO_{2-x} towards dry reforming of methaneREVIEWER COMMENTS

Reviewer #1 (Remarks to the Author):

Hui Wang and colleagues investigated the synergistic catalysis at Ir@CeO_{2-x} interface for the dry reforming of methane reaction. The manuscript is written in a concise manner and the claims are mostly supported by the experimental data and theoretical calculations. The developed materials indeed exhibit extraordinarily high catalytic activity. The novelty and the unique character of the Ir metal-ceria interface are exaggerated, especially since there are numerous studies showing the substantially improved catalysis over metal-ceria interfaces for several metals (Pt, Ni, Cu.) and catalytic reactions. This work can be published after the following issues are addressed:

1. I am not convinced that ceria nanorods are square in their cross section and expose a similar fraction of 110 and 111 facets as suggested in figure 1d and supplementary figures 3-8. Their shape and dominant exposed facets depends on the digestion time, temperature and alkali concentration. See for example <https://doi.org/10.1021/acsami.2c05221>. A more thorough analysis is required to identify the actual facets of ceria.
2. Also, if both 111 and 110 are present and catalytic activity is a sum of their contributions, why was all the DFT done only on one ceria facet? Could DFT calculations on 111 facet bring the estimated activation barriers closer to experimental values?
3. DFT calculations should be added also for the stoichiometric Ir/CeO₂ interface without the oxygen vacancy and for the extended Ir terrace to emphasize the benefit of oxygen deficient ceria.
4. What is the actual oxidation state of interfacial iridium in the working catalyst? Current version of the manuscript does not provide conclusive answers. XANES fitting was done with IrO₂, whereas XPS suggests a more electron rich state of Ir (Ir^{δ+}). Could other references for cationic iridium produce a similarly good fit in XANES?
5. Pulse chemisorption of CO to estimate Ir dispersion on activated catalysts should be verified with DRIFT spectroscopy to exclude carbonate formation on the partly reduced ceria. This is what usually happens and would overestimate the dispersion values.
6. The colour map 2D images for DRIFT spectroscopy are attractive to look at, but crucial information can be overlooked due to low colour resolution. I suggest individual spectra are compared at suitable time intervals to maximize the spectroscopic signal information and compare those. 2D images can still be kept in ESI.

Reviewer #2 (Remarks to the Author):

The paper by Wang et al. is a comprehensive study of Ir/CeO₂ catalysts for the dry reforming of methane. My observations are reported below.

- 1) In my opinion, XPS results need some more experimental detail/support. The main issue I see is related to sample manipulation after the pretreatment and prior to the transfer to the XPS apparatus. Indeed, the characterization, as described also in the experimental part, appears to be carried out fully ex-situ, which can be fine but the authors would need to provide evidence of the reliability of their procedure in representing the "operating" state of their catalyst.
- 2) In their discussion of Ir-Ce SMSI the authors should consider the work by Li et al. (ref. 29 of the manuscript), that has been cited incidentally in the introduction but not further discussed. The tested reactions are different (DRM in this paper, CO₂ hydrogenation in the paper by Li), but some of the general outcomes that are not directly related to the reaction atmosphere are similar, particularly regarding the effect of Ir loading on Ir oxidation state and the concomitant SMSI. In my opinion, this somehow limits the novelty of this work.
- 3) I would not use the ~ symbol before the energies of XPS peaks. They should be univocally defined.
- 4) English needs revision.
- 5) Page 11, lines 201-202: a reference to the calculation of the Mears criteria in the Supplementary Information should be included.

Reviewer #3 (Remarks to the Author):

A very interesting study involving Ir-based catalysts for methane dry reforming. The work could have a strong impact on the topic under study, but several key points need clarification or additional research.

- 1) Top of page 5: Ir cations can activate methane at moderate temperatures (< 500 K). see: Salvatore et al, ACS Appl. Nano Mater. 2021, 4, 11146. Thus, methane activation is not always the rate determining step. This needs to be mentioned.
- 2) Top of page 8: The stabilization of Ce³⁺ could help methane and CO₂ activation and facilitate the dry reforming process (see cited refs 21 and 24).
- 3) Top of page 9, explain the exact meaning of "quasi in situ XPS".
- 4) Top of page 11, "reported dates tested"??
- 5) Pages 14 and 15: After examining the discussed data, it is not clear that the activation of CH₄ is always more difficult than the activation of CO₂.
- 6) Pages 17 and 18: The model used for the theoretical calculations needs a better justification. It is not clear that it truly represents the samples used in the experimental part.

Response to Reviewers

Reviewer #1

Comments: Hui Wang and colleagues investigated the synergistic catalysis at Ir@CeO_{2-x} interface for the dry reforming of methane reaction. The manuscript is written in a concise manner and the claims are mostly supported by the experimental data and theoretical calculations. The developed materials indeed exhibit extraordinarily high catalytic activity. The novelty and the unique character of the Ir metal-ceria interface are exaggerated, especially since there are numerous studies showing the substantially improved catalysis over metal-ceria interfaces for several metals (Pt, Ni, Cu.) and catalytic reactions. This work can be published after the following issues are addressed:

(1) I am not convinced that ceria nanorods are square in their cross section and expose a similar fraction of 110 and 111 facets as suggested in figure 1d and supplementary figures 3-8. Their shape and dominant exposed facets depends on the digestion time, temperature and alkali concentration. See for example <https://doi.org/10.1021/acsami.2c05221>. A more thorough analysis is required to identify the actual facets of ceria.

Author reply: Thank you for this comment. For the CeO₂ support, we agree with the reviewer “their shape and dominant exposed facets depends on the digestion time, temperature and alkali concentration”. In the original manuscript, in order to clearly observe the exposed crystal facets of CeO₂ support, Fig. 1d and Supplementary Fig. 3–7 supply the local magnification HR-TEM images of various samples with different loadings, which might lead to some misunderstanding that “ceria nanorods are square in their cross section and expose a similar fraction of (110) and (111) facets”. According to this comment, we carefully re-analyzed the HR-TEM results. As shown in Supplementary Fig. 8, the (110) facet is predominantly exposed accompanied with minor (111) facet for the CeO₂ nanorods support, consistent with the results reported in the literature (*ACS Catal.* 2020, 10, 11493; *ACS Catal.* 2020, 10, 4003; *Angew. Chem. Int. Ed.* 2023, 62, e202305661).

• **Page 6, Line 3: rephrase:** “From the local magnification HR-TEM images (Fig. 1d and Supplementary Fig. 3–7), two clear crystalline phases are identified as ~0.19 and ~0.31 nm, respectively, corresponding to the (110) and (111) planes of CeO₂ nanorods support. As shown in Supplementary Fig. 8, the (110) facet is predominantly exposed accompanied with minor (111) facet, consistent with the previous reports^{43,44}.”

• **Supplementary Figure 8** has been added in the revised Supplementary Information.

Supplementary Figure 8. a–d HR-TEM images of various Ir/CeO_{2-x} samples with Ir loading of 0.2%, 0.6%, 2.0% and 3.0%, respectively. The yellow and green borders represent the exposed (110) and (111) crystal facets of CeO₂ nanorods, respectively.

(2) Also, if both 111 and 110 are present and catalytic activity is a sum of their contributions, why was all the DFT done only on one ceria facet? Could DFT calculations on 111 facet bring the estimated activation barriers closer to experimental values?

Author reply: Thank you for this comment. In order to investigate the contribution of the (111)

crystal plane to catalytic activity, we prepared 0.6% Ir/CeO₂ octahedrons sample with only (111) facet exposed (Fig. R1) and evaluated its catalytic performance towards DRM reaction under the same reaction conditions as 0.6% Ir/CeO₂ nanorods sample. As shown in Supplementary Fig. 21, both the CH₄ and CO₂ conversion are quite low. Therefore, we believe that the contribution of the (111) crystal plane is very small to the overall catalytic activity of 0.6% Ir/CeO_{2-x} nanorods catalyst. In addition, the relative ratios of both surface Ce³⁺ and oxygen vacancy (O_v) for Ir/CeO₂ octahedrons sample are much lower than those of Ir/CeO₂ nanorods sample according to the *quasi in situ* XPS results (Supplementary Fig. 15), which may limit the activation of CO₂. The results in this work are consistent with the previously reported experimental and theoretical studies (*ACS Catal.* 2020, 10, 613; *Surf. Sci.* 2005, 595, 223).

According to this comment, we also built the Ir/CeO₂ (111) model and calculated the full potential reaction pathway of CH₄ decomposition (Supplementary Fig. 39 and 40) in the revised Supplementary Information. The reaction energy barrier of rate-determining step in the Ir/CeO₂ (111) model (2.25 eV) is significantly larger than in the Ir/CeO₂ (110) one (Fig. 5: 1.43 eV), in well agreement with the catalytic performance in Supplementary Fig. 21. Therefore, the catalytic activity is primarily contributed by the (110) crystal plane.

Figure R1. **a** TEM and **b** HR-TEM images of the 0.6% Ir/CeO₂ octahedrons sample. The inset in **b** shows the corresponding FFT image.

- **Supplementary Figures 15, 21, 39 and 40** have been supplemented in the revised Supplementary Information.

Supplementary Figure 15. *Quasi in situ* XPS of **a** Ce 3d and **b** O 1s for the 0.6% Ir/CeO_{2-x} nanorods and 0.6% Ir/CeO_{2-x} octahedrons samples, respectively.

Supplementary Figure 21. CH₄ conversion and CO₂ conversion over the 0.6% Ir/CeO_{2-x} nanorods and 0.6% Ir/CeO_{2-x} octahedrons samples at 650, 700 and 750 °C, respectively.

Supplementary Figure 39. Potential energy profile for CH₄ decomposition on the surface of

Ir/CeO₂ (111) based on DFT calculations. ‘TS’ denotes a transition state. The black and orange numbers represent the adsorption energy and energy barrier of elementary steps, respectively.

Supplementary Figure 40. Schematic illustration for the CH₄ decomposition on Ir/CeO₂ (111). Ir, green; Ce, yellow; C, grey; O, crimson; H, white.

(3) DFT calculations should be added also for the stoichiometric Ir/CeO₂ interface without the oxygen vacancy and for the extended Ir terrace to emphasize the benefit of oxygen deficient ceria.

Author reply: This is a valuable suggestion to improve the manuscript. According to this comment, we built the Ir/CeO₂ (110) model without the oxygen vacancy and calculated the full potential reaction pathway of DRM. As shown in Supplementary Fig. 41, the dehydrogenation of CH₄* to CH₃* gives the highest energy barrier (1.90 eV) and thus is determined as the rate-determining step in this case. This is higher than the Ir/CeO_{2-x} (110) (Fig. 5: 1.43 eV) model. In Supplementary Fig. 42, CO₂ is adsorbed and decomposed on Ir cluster in the case of Ir/CeO₂ (110) model, in contrast to the Ir/CeO_{2-x} (110) model where oxygen vacancy serves as the active site. Moreover, the lack of oxygen vacancy leads to a much higher energy barrier of CO generation in Ir/CeO₂ (110) (Supplementary Fig. 41: 1.65 eV) than that in Ir/CeO_{2-x} (110) (Fig. 5: 0.73 eV). The calculation results confirm the significant role of interface structure (Ir^{δ+}-O_v-Ce³⁺) in Ir/CeO_{2-x} (110) catalyst toward DRM reaction.

● **Supplementary Figures 41 and 42** have been added in the revised Supplementary Information.

Supplementary Figure 41. Potential energy profile for CH₄ and CO₂ decomposition on Ir/CeO₂ (110) without oxygen vacancy based on DFT calculations. ‘TS’ denotes a transition state. The black and orange numbers represent the adsorption energy and energy barrier of elementary steps, respectively.

Supplementary Figure 42. Schematic illustration for CH₄ and CO₂ decomposition on Ir/CeO₂ (110) without oxygen vacancy. Ir, green; Ce, yellow; C, grey; O, crimson; H, white.

(4) What is the actual oxidation state of interfacial iridium in the working catalyst? Current version of the manuscript does not provide conclusive answers. XANES fitting was done with IrO₂, whereas XPS suggests a more electron rich state of Ir (Ir^{δ+}). Could other references for cationic iridium produce a similarly good fit in XANES?

Author reply: Thank you for this comment. According to this comment, we carried out the linear

combination fitting (LCF) analysis on the XANES data to obtain the average oxidation state of iridium species and the results were shown in Supplementary Table 4. The average valence state of iridium species displays the following sequence: 0.2% Ir/CeO_{2-x} (+2.5) > 0.6% Ir/CeO_{2-x} (+2.4) > 2.0% Ir/CeO_{2-x} (+1.8) > 3.0% Ir/CeO_{2-x} (+1.4) > 0.6% Ir/Al₂O₃ (+1.1). Corresponding discussion has been added in the revised manuscript.

Based on this comment, we conducted a comprehensive survey of the published literature about XANES studies on iridium species (*Nat. Commun.* 2022, 13, 7754; *J. Am. Chem. Soc.* 2023, 145, 6658; *Angew. Chem. Int. Ed.* 2023, 62, e202310973; *J. Am. Chem. Soc.* 2020, 142, 18378; *Nat. Commun.* 2021, 12, 6118; *ACS Catal.* 2023, 13, 12153), where Ir foil and IrO₂ have been commonly used as reference samples but other cationic iridium has not been reported.

- **Page 9, Line 4: rephrase:** “The average oxidation state of iridium species is calculated based on the results from LCF analysis, which gives the following sequence: 0.2% Ir/CeO_{2-x} (+2.5) > 0.6% Ir/CeO_{2-x} (+2.4) > 2.0% Ir/CeO_{2-x} (+1.8) > 3.0% Ir/CeO_{2-x} (+1.7) > 0.6% Ir/Al₂O₃ (+1.1).”
- **Supplementary Table 4** has been improved in the revised Supplementary Information.

Supplementary Table 4. LCF fitting results of the Ir L₃-edge for various samples

Sample	Ir ⁴⁺ atomic ratio	Ir ⁰ atomic ratio	Average valence state of Ir species
0.6%Ir/Al ₂ O ₃	26.9%	73.1%	+1.1
0.2% Ir/CeO _{2-x}	63.2%	36.8%	+2.5
0.6% Ir/CeO _{2-x}	59.2%	40.8%	+2.4
2.0% Ir/CeO _{2-x}	46.2%	53.8%	+1.8
3.0% Ir/CeO _{2-x}	43.0%	57.0%	+1.7

(5) Pulse chemisorption of CO to estimate Ir dispersion on activated catalysts should be verified with DRIFT spectroscopy to exclude carbonate formation on the partly reduced ceria. This is what usually happens and would overestimate the dispersion values.

Author reply: Thank you for this valuable comment. We apologize for ignoring the adsorption of

CO on Ce^{3+} . According to this comment, we performed *in situ* CO-DRIFTS chemisorption experiment on CeO_2 nanorods sample. After the same pretreatment process as Ir/ CeO_{2-x} samples, the CeO_2 nanorods sample was partially reduced with the formation of Ce^{3+} (Supplementary Fig. 13). As shown in Fig. R2, the signal of carbonate is detected, indicating the consumption of CO by partially reduced CeO_2 . Therefore, we made a correction for CO deduction in the CO pulse chemisorption experiments, where the CO consumption on partially reduced ceria in Ir/ CeO_{2-x} samples is obtained by means of respective Ce^{3+} ratio from *quasi in situ* XPS results. The corrected Ir dispersion is displayed in Supplementary Table 1, and corresponding information and discussion have been improved in the revised manuscript and Supplementary Information.

• **Page 6, Line 2: rephrase:** “Accordingly, the dispersion of Ir (D_{Ir} , Supplementary Table 1) decreases gradually from 82% (0.2% Ir/ CeO_{2-x}) to 39% (3.0% Ir/ CeO_{2-x}).”

• **Page 12, Line 9: rephrase:** “In addition, the intrinsic TOF value of CH_4 is evaluated at a low conversion (below 15%, Fig. 3h), which gives a decrease order as follows: 0.2% Ir/ CeO_{2-x} ($168 \text{ mol}_{\text{CH}_4} \text{ mol}_{\text{Ir}}^{-1} \text{ s}^{-1}$) > 0.6% Ir/ CeO_{2-x} ($163 \text{ mol}_{\text{CH}_4} \text{ mol}_{\text{Ir}}^{-1} \text{ s}^{-1}$) > 2.0% Ir/ CeO_{2-x} ($122 \text{ mol}_{\text{CH}_4} \text{ mol}_{\text{Ir}}^{-1} \text{ s}^{-1}$) > 3.0% Ir/ CeO_{2-x} ($110 \text{ mol}_{\text{CH}_4} \text{ mol}_{\text{Ir}}^{-1} \text{ s}^{-1}$).”

Figure R2. *In situ* CO-DRIFTS spectra over CeO_2 nanorods support and various Ir/ CeO_{2-x} samples.

• **Figure 3h and i** have been corrected in the revised manuscript.

Fig. 3 Catalytic performance of various catalysts. **a, b** CH₄ conversion and CO₂ conversion over various samples at 650, 700 and 750 °C. **c** Comparison study on mass specific activity between 0.6% Ir/CeO_{2-x} and other typical catalysts used in DRM reaction. **d** Stability test of 0.6% Ir/CeO_{2-x} and 0.6% Ir/Al₂O₃ at 700 °C for DRM. Evaluated conditions: CH₄/CO₂/N₂ = 20/20/5 mL min⁻¹, GHSV=240000 mL g⁻¹ h⁻¹. **e, f** Correlation of CH₄ or CO₂ partial pressure on the reaction rates of CH₄ and CO₂. **g.** Kinetic studies and calculated activation energy (E_a) of CH₄ over various catalysts. **h** Intrinsic TOF over various catalysts within the catalytic dynamic range. **i** TOF as a function of interfacial Ir concentration calculated by CO-DRIFTS results.

• **Supplementary Table 1** has been improved in the revised Supplementary Information.

Supplementary Table 1. Physicochemical properties of various samples

Sample	Ir content ^a (wt. %)	BET Surface Area ^b (m ² g ⁻¹)	Mean pore size ^b (nm)	Mean Ir particle size ^c (nm)	Theoretical dispersion of Ir ^d (%)	Determined dispersion of Ir ^e (%)
0.6%Ir/Al ₂ O ₃	0.6	/	/	1.3	76.9	65.2
0.2% Ir/CeO _{2-x}	0.2	91.9	22.6	1.0	99.0	81.9
0.6% Ir/CeO _{2-x}	0.6	87.9	22.4	1.0	98.0	79.1
2.0% Ir/CeO _{2-x}	2.0	85.2	21.2	1.5	66.7	45.7
3.0% Ir/CeO _{2-x}	3.0	83.6	20.4	1.6	62.5	38.9

^a Ir loading was calculated based on ICP-AES results.

^b Specific surface area and mean pore size were determined by BET measurements.

^c The mean Ir size was determined by TEM images.

^d Theoretical dispersion of Ir was estimated on the basis of Ir particle size from TEM (assuming Ir as a spherical particle).

^e Determined dispersion of Ir was obtained from CO pulse chemisorption experiments results.

- **Supplementary Figures 24 and 25** have been corrected in the revised Supplementary Information.

Supplementary Figure 24. TOF value as a function of surface concentration of Ce³⁺/(Ce³⁺ + Ce⁴⁺) ratio calculated by *quasi in situ* XPS results.

Supplementary Figure 25. TOF value as a function of surface oxygen vacancy ratio calculated by *quasi in situ* XPS results.

(6) The colour map 2D images for DRIFT spectroscopy are attractive to look at, but crucial information can be overlooked due to low colour resolution. I suggest individual spectra are compared at suitable time intervals to maximize the spectroscopic signal information and compare those. 2D images can still be kept in ESI.

Author reply: This is a valuable suggestion to improve the manuscript. According to this comment, we supplemented the line chart of *in situ/operando* DRIFTS spectra on the 0.6% Ir/CeO_{2-x} sample in the revised Supplementary Information.

• **Supplementary Figure 28** has been added in the revised Supplementary Information.

Supplementary Figure 28. *In situ/operando* DRIFTS spectra over 0.6% Ir/CeO_{2-x} sample at 700 °C after *in-situ* pretreatment and He purging, followed by exposure to: **a–d** first CH₄ atmosphere, subsequent CH₄+CO₂ and then CH₄ atmosphere for 30 min, respectively; **e–h** first CO₂ atmosphere, subsequent CO₂+CH₄ and then CO₂ atmosphere for 30 min, respectively.

Reviewer #2

Comments: The paper by Wang et al. is a comprehensive study of Ir/CeO₂ catalysts for the dry reforming of methane. My observations are reported below.

(1) In my opinion, XPS results need some more experimental detail/support. The main issue I see is related to sample manipulation after the pretreatment and prior to the transfer to the XPS apparatus. Indeed, the characterization, as described also in the experimental part, appears to be carried out fully ex-situ, which can be fine but the authors would need to provide evidence of the reliability of their procedure in representing the "operating" state of their catalyst.

Author reply: Thank you for this comment. According to this comment, the detailed procedure of sample handling has been described in the revised Supplementary Information.

• **Supplementary Information, Page 3, Line 11: rephrase:** “For the XPS testing samples, air is strictly isolated during sample pretreatment and transfer. Typically, the sample was firstly held by quartz wool and was placed in the middle of a quartz tube reactor. After the pre-treated process, the sample was cooled down to room temperature in N₂ flowing. Then, the reactor was sealed and carefully transferred into a N₂ atmosphere glove box, followed by sample preparation and installation in an airtight transport chamber for XPS measurement.”

(2) In their discussion of Ir-Ce SMSI the authors should consider the work by Li et al. (ref. 29 of the manuscript), that has been cited incidentally in the introduction but not further discussed. The tested reactions are different (DRM in this paper, CO₂ hydrogenation in the paper by Li), but some of the general outcomes that are not directly related to the reaction

atmosphere are similar, particularly regarding the effect of Ir loading on Ir oxidation state and the concomitant SMSI. In my opinion, this somehow limits the novelty of this work.

Author reply: Thank you for this comment. The work of Li et al. reported a series of Ir/CeO₂ nanocatalysts with various Ir loadings (from 5% to 20%) and evaluated their catalytic performance towards CO₂ hydrogenation reaction. The results show that the chemical state of Ir species determined by its loading (5%: partially oxidized Ir species; 20%: metallic Ir) has a major impact on the reaction selectivity. In this work, we report a sub-nanometric Ir clusters anchored onto CeO₂ nanorods catalyst, which shows excellent catalytic performance towards DRM reaction. A comprehensive investigation including AC-HAADF-STEM, *quasi in situ* XPS and *in situ* XAFS confirms the formation of a unique metal-support interface structure (Ir^{δ+}-O_v-Ce³⁺; O_v denotes oxygen vacancy), whose concentration can be modulated facilely *via* changing Ir loading (from 0.2% to 3%). This interfacial synergistic catalysis resulting from SMSI is demonstrated based on experimental investigations and DFT calculations, which not only promotes the catalytic activity for DRM reaction, but also inhibits catalyst deactivation from coke formation. This work demonstrates a new strategy for rational design and preparation of highly-efficient catalysts towards activation of both C–H and C=O bonds, and provides atomic-level insights into interfacial synergistic catalysis for DRM reaction.

(3) I would not use the ~ symbol before the energies of XPS peaks. They should be univocally defined.

Author reply: Thank you for this comment. According to this comment, we have precisely defined the location of XPS peaks and corrected this description in the revised manuscript.

(4) English needs revision.

Author reply: Thank you for this comment. We have checked carefully and polished English writing throughout the manuscript.

(5) Page 11, lines 201-202: a reference to the calculation of the Mears criteria in the Supplementary Information should be included.

Author reply: Thank you for this comment. The related references about the Mears criteria calculation have been cited in the revised manuscript (refs. 35, 52 and 53) and the revised Supplementary Information (refs. 1, 2 and 3).

• **Page 11, Line 17: rephrase:** “Firstly, the effects of external and internal diffusion limitation have been eliminated under the aforementioned reaction conditions^{35,52,53}.”

Reference

35. Meng, H. et al. A strong bimetal-support interaction in ethanol steam reforming. *Nat. Commun.* **14**, 3189 (2023).

52. Wang, J. et al. Design of a carbon-resistant Ni@S-2 reforming catalyst: controllable Ni nanoparticles sandwiched in a peasecod-like structure. *Appl. Catal. B Environ.* **282**, 119546 (2021).

53. Yan, X. et al. Highly efficient and stable Ni/CeO₂-SiO₂ catalyst for dry reforming of methane: effect of interfacial structure of Ni/CeO₂ on SiO₂. *Appl. Catal. B Environ.* **246**, 221–231 (2019).

• **Supplementary Information, Page 4, Line 21: rephrase:** “Mears criterion for external

diffusion. If $\frac{-r_A' \rho_b Rn}{k_c C_{Ab}} < 0.15$, then the external mass transfer effects can be neglected¹⁻³.”

Reviewer #3

Comments: A very interesting study involving Ir-based catalysts for methane dry reforming. The work could have a strong impact on the topic under study, but several key points need clarification or additional research.

(1) Top of page 5: Ir cations can activate methane at moderate temperatures (< 500 K). see: Salvatore et al, ACS Appl. Nano Mater. 2021, 4, 11146. Thus, methane activation is not always the rate determining step. This needs to be mentioned.

Author reply: Thank you for this comment. It has been reported that C–H can be activated at moderate temperatures (< 500 K) when only CH₄ was introduced in the presence of Ni/CeO₂ or Cu@IrO₂ catalyst (*ACS Catal.* 2016, 6, 8184; *J. Am. Chem. Soc.* 2019, 141, 5303; *Angew. Chem.*

Int. Ed. 2017, 129,13221; *ACS Catal.* 2022, 12, 5470). However, in the case of DRM reaction, the CH₄ dissociation to C (~1662 kJ mol⁻¹) is more difficult than that of CO₂ dissociation (~750 kJ mol⁻¹) from thermodynamic analysis. The dissociation of CH₄ is normally confirmed as the rate-determining step based on both experimental studies and DFT calculations (*J. Am. Chem. Soc.* 2017, 139, 1937; *Nat. Commun.* 2019, 10, 5181; *Appl. Catal. B* 2019, 246, 221; *Chem Catal.* 2022, 2, 1748; *Chem. Soc. Rev.* 2014, 43, 7813).

In this work, based on the kinetic experimental data, the calculated reaction order with respect to CH₄ (~0.67 and ~0.53) is significantly higher than that of CO₂ (~0.09 and ~0.07), indicating that the CH₄ activation is critical to the reaction kinetics. Moreover, the apparent activation energy (E_a) of CH₄ over 0.6% Ir/CeO_{2-x} is 91 kJ mol⁻¹, much larger than that of CO₂ (70 kJ mol⁻¹) (Fig. 3g and Supplementary Fig. 22 and 23). In addition, according to the DFT calculation results, the energy barrier of CH₄ dissociation (1.43 eV, Fig. 5i) is significantly larger than that of CO₂ dissociation (0.70 eV, Supplementary Figs. 33 and 34). The results verify that the CH₄ dissociation entails a higher energy barrier and serves as the rate-determining step in this catalytic system. In summary, the CH₄ activation at moderate temperatures does not conflict with the conclusion that CH₄ dissociation is the rate-determining step in DRM reaction, since the former only involves the first C–H bond dissociation whilst the latter includes the dissociation of all the four C–H bonds and the oxidation of C to CO.

(2) Top of page 8: The stabilization of Ce³⁺ could help methane and CO₂ activation and facilitate the dry reforming process (see cited refs 21 and 24).

Author reply: Thank you for this comment. In this work, we found a similar conclusion with the reference that “a partially oxidized state of ruthenium stabilized by reduced ceria (Ru^{δ+}–CeO_{2-x}) to sustain active chemistry”. Specifically, the strong electronic metal-support interaction between Ir species and CeO₂ stabilizes the Ir^{δ+} species and facilitates the formation of interfacial structure (Ir^{δ+}–O_v–Ce³⁺). Both experimental investigations (catalytic evaluations, *operando* DRIFTS and *operando* XAFS) and DFT calculations substantiate the Ir^{δ+}–O_v–Ce³⁺ structure serves as active site towards DRM reaction: CH₄ molecule experiences activation adsorption and dissociation to CH₂* species and H₂ at Ir^{δ+} species; whilst CO₂ molecule undergoes activation adsorption at

O_v-Ce³⁺ sites (O_v and Ce³⁺ are concomitant).

(3) Top of page 9, explain the exact meaning of "quasi in situ XPS".

Author reply: Thank you for this comment. The detailed procedure of sample handling has been described in the revised Supplementary Information.

• **Supplementary Information, Page 3, Line 11: rephrase:** “For the XPS testing samples, air is strictly isolated during sample pretreatment and transfer. Typically, the sample was firstly held by quartz wool and was placed in the middle of a quartz tube reactor. After the pre-treated process, the sample was cooled down to room temperature in N₂ flowing. Then, the reactor was sealed and carefully transferred into a N₂ atmosphere glove box, followed by sample preparation and installation in an airtight transport chamber for XPS measurement.”

(4) Top of page 11, "reported dates tested"?

Author reply: We apologize for this spelling mistake. This has been corrected in the revised manuscript.

• **Page 11, Line 3: rephrase:** “As for the optimal 0.6% Ir/CeO_{2-x} sample, both the CH₄ and CO₂ conversions reach up to the thermodynamic equilibrium, and the reaction rate is 3–20 times higher than previously reported studies under similar reaction conditions within 650–750 °C (Fig. 3c and Supplementary Table 5)^{20,52}.”

(5) Pages 14 and 15: After examining the discussed data, it is not clear that the activation of CH₄ is always more difficult than the activation of CO₂.

Author reply: Thank you for this valuable comment. From thermodynamic analysis, the CH₄ dissociation to C (~1662 kJ mol⁻¹) is more difficult than that of CO₂ dissociation (~750 kJ mol⁻¹). In this work, based on the kinetic experimental data, the calculated reaction order with respect to CH₄ (~0.67 and ~0.53) is significantly higher than that of CO₂ (~0.09 and ~0.07), indicating that the CH₄ activation is critical to the reaction kinetics. Moreover, the apparent activation energy (*E_a*) of CH₄ over 0.6% Ir/CeO_{2-x} is 91 kJ mol⁻¹, much larger than that of CO₂ (70 kJ mol⁻¹) (Fig.

3g and Supplementary Fig. 22 and 23). In addition, according to the DFT calculation results, the energy barrier of CH₄ dissociation (1.43 eV, Fig. 5i) is significantly larger than that of CO₂ dissociation (0.70 eV, Supplementary Fig. 33 and 34). The results verify that the CH₄ dissociation entails a higher energy barrier and serves as the rate-determining step in this catalytic system, which is consistent with previously reported studies (*J. Am. Chem. Soc.* 2017, 139, 1937; *Nat. Commun.* 2019, 10, 5181; *Appl. Catal. B* 2019, 246, 221; *Chem Catal.* 2022, 2, 1748; *Chem. Soc. Rev.* 2014, 43, 7813).

● **Page 18, Line 17: rephrase:** “According to the calculation results, the dehydrogenation of CH₃* species to CH₂* gives the highest energy barrier (1.43 eV), which is determined as the rate-determining step of DRM reaction, in accordance with the experimental results (Fig. 3e–f).”

(6) Pages 17 and 18: The model used for the theoretical calculations needs a better justification. It is not clear that it truly represents the samples used in the experimental part.

Author reply: Thank you for this comment. We established and optimized the Ir nanocluster (seven Ir atoms) supported on CeO_{2-x} containing O_v, based on the AC-HAADF-STEM and electronic structure characterizations (Fig. 2a–g), so as to represent the interface structure (Ir^{δ+}–O_v–Ce³⁺). During the calculation process of CH₄ oxidation to CH₂O* and CO₂* decomposition to CO*, the variation in charge transfer is consistent with *in situ* normalized XANES and *quasi in situ* XPS results, and the key reaction intermediate (CH₂O*) accords with the signals observed by *in situ/operando* DRIFTS spectra.

According to this comment, we built the Ir/CeO₂ (110) model without the oxygen vacancy and calculated the full potential reaction pathway of DRM, so as to perform a comparative study. As shown in Supplementary Fig. 41, the dehydrogenation of CH₄* to CH₃* gives the highest energy barrier (1.90 eV) and thus is determined as the rate-determining step in this case. This is higher than the Ir/CeO_{2-x} (110) (Fig. 5: 1.43 eV) model. In Supplementary Fig. 42, CO₂ is adsorbed and decomposed on Ir cluster in the case of Ir/CeO₂ (110) model, in contrast to the Ir/CeO_{2-x} (110) model where oxygen vacancy serves as the active site. Moreover, the lack of oxygen vacancy leads to a much higher energy barrier of CO generation in Ir/CeO₂ (110)

(Supplementary Fig. 41: 1.65 eV) than that in Ir/CeO_{2-x}(110) (Fig. 5: 0.73 eV). The calculation results confirm the significant role of interface structure (Ir^{δ+}-O_v-Ce³⁺) in Ir/CeO_{2-x}(110) catalyst toward DRM reaction.

• **Supplementary Figure 41 and 42** have been added in the revised Supplementary Information.

Supplementary Figure 41. Potential energy profile for CH₄ and CO₂ decomposition on Ir/CeO₂(110) without oxygen vacancy based on DFT calculations. ‘TS’ denotes a transition state. The black and orange numbers represent the adsorption energy and energy barrier of elementary steps, respectively.

Supplementary Figure 42. Schematic illustration for CH₄ and CO₂ decomposition on Ir/CeO₂(110) without oxygen vacancy. Ir, green; Ce, yellow; C, grey; O, crimson; H, white.

REVIEWERS' COMMENTS

Reviewer #1 (Remarks to the Author):

The manuscript was thoroughly revised and all my questions and comments were answered properly. This work can now be published.

Reviewer #2 (Remarks to the Author):

The authors addressed satisfactorily the issues raised by the reviewers. The paper can be published in its present form.

Reviewer #3 (Remarks to the Author):

The response of the authors to my previous comments is satisfactory. The authors have really improved the level of the study and I recommend acceptance for publication.

Response to Reviewers

Reviewer #1

Comments:

The manuscript was thoroughly revised and all my questions and comments were answered properly. This work can now be published.

Author reply: Thank you for this comment.

Reviewer #2

Comments:

The authors addressed satisfactorily the issues raised by the reviewers. The paper can be published in its present form.

Author reply: Thank you for this comment.

Reviewer #3

Comments:

The response of the authors to my previous comments is satisfactory. The authors have really improved the level of the study and I recommend acceptance for publication.

Author reply: Thank you for this comment.